# *In-vitro* evaluation of probiotic potential of gut microbes isolated from retail chicken

Sangram Biswas[1], Lutfor Rahman[2], Md. Taofiqur Rahman[1], Susmita Chowdhury[3], Fahmida Khatun[1], Azimun Nahar[4], Sabina Yasmin[1]*

1 Department of Biotechnology, Bangladesh Agricultural University (BAU), Mymensingh, Bangladesh, 2 Department of Microbiology and Hygiene, Bangladesh Agricultural University (BAU), Mymensingh, Bangladesh, 3 Department of Dietetics and Nutrition, Florida International University (FIU), Florida, United States of America, 4 Department of Medicine, Bangladesh Agricultural University (BAU), Mymensingh, Bangladesh

* sabina.yasmin@bau.edu.bd

## Abstract

Probiotics are live, non-pathogenic microorganisms that help to improve the host's gut health when administrated in sufficient proportions and are now serving as effective alternatives to antibiotics for managing animal infections and enhancing production. The objective of this study was to isolate, identify and characterize lactic acid bacteria (LAB) strains with excellent probiotic properties from the gastrointestinal tract (GIT) of retail broiler chickens. Samples were enriched in MRS broth at 37°C and plated on MRS agar to isolate distinct colonies of potential probiotic candidates. The isolates underwent a series of standard morphological and biochemical analysis to fulfill the criteria for presumptive identification of LAB and probiotic characteristics. These analyses included Gram staining, catalase testing, hemolytic activity assays, tolerance assays to NaCl, simulated gastric juice and bile salts, antagonistic activity assays, antibiotic susceptibility testing, cell adhesion assay and genotypic identification through 16S rRNA gene sequencing. A total of 40 microbial strains were isolated from the GIT of 20 retail broiler chickens. Among these, 4 LAB strains showed the best probiotic results and were genotypically identified as *Enterococcus faecium* MCI7, *Pedicoccus pentosaceus* MCI10, *Pediococcus pentosaceus* MCC6 and *Pediococcus pentosaceus* MCC12. The selected strains exhibited non-hemolytic activity and were able to survive in simulated gastric juice at pH 3. Furthermore, the strains displayed bile salt tolerance in the presence of 0.3% bile salt for 4 hours, ranging from 21.91 to 32.77% and a wide range of antimicrobial activities against various pathogenic bacterial strains with inhibition zones ranging from 10 to 16.5 mm. Moreover, three *P. pentosaceus* strains (MCI10, MCC6, MCC12) were sensitive to most of the tested antibiotics and demonstrated good adherence abilities. Our study identified four LAB strains as promising probiotic candidates for poultry

**Data availability statement:** All relevant data are within the manuscript and its Supporting Information files.

**Funding:** The author(s) received no specific funding for this work.

**Competing interests:** The authors have declared that no competing interests exist.

feed additives to effectively establish intestinal microflora, enhance meat quality and growth, and control pathogens.

---

## 1. Introduction

The poultry industry significantly contributes to economic growth through job creation and trade, and helps to meet global nutritional needs by providing accessible protein and essential micronutrients [1]. The Food and Agriculture Organization (FAO) and Economic Co-operation and Development (OECD) expect that consumption of poultry meat globally will increase more than 16% and reach 160 million metric tons by 2033 [2]. The global population is projected to reach 9.7 billion by 2050, with global meat consumption expected to increase by 80% due to rising demand for meat protein [3]. In Bangladesh, the annual poultry consumption per capita in 2025 is expected to reach approximately 1.44 kilograms [4]. The poultry production significantly contributes to the national economy of Bangladesh, involving around 1 million entrepreneurs and employing 8 million people [5]. Additionally, this sector has significantly contributed to the nation's advancement of the Sustainable Development Goals (SDGs) by addressing malnutrition and promoting better health by providing protein-rich foods [6]. Therefore, safeguarding the growth of the poultry industry is thought to be crucial to improving the nutritional status and economic development of the Bangladeshi people.

However, the biggest challenges for poultry farming are lack of experience in managing farms, inadequate management practices, insufficient disease control programs, and improper vaccination protocols [7]. Small-scale poultry farming in Bangladesh poses significant public and poultry health risks due to close human-poultry contact and a lack of biosecurity measures [5]. Additionally, farmers use antibiotics to make them grow faster and prevent diseases [8]. The therapeutic use of antibiotics in poultry feed raises significant concerns about the presence of antibiotic residues in poultry products, which can pose risks to consumers [9].

Antimicrobial resistance (AMR) is a natural and ancient phenomenon in bacteria, but human activities, such as the uncontrolled use of antimicrobial agents, have accelerated its emergence and spread. The food animal system becomes a repository for antibiotic-resistant genes as a result of the overuse of antibiotics, which directly selects resistant microorganisms [10,11]. Therefore, many European Union (EU) countries banned the marketing and use of antimicrobial growth promoters (AGPs) in feed nutrition since 2006 due to their harmful effects, including the development of antibiotic resistance, damaging the natural gut flora, and increased risk of infections [12]. Following the ban on antimicrobial growth promoters (AGPs) and increasing consumer pressure, food animal producers and scientists explore alternatives like probiotics, prebiotics, organic acids, essential oils, phytochemicals, in-feed enzymes and so on [13,14]. Among these alternatives, probiotics are often preferred due to their ability to directly introduce live beneficial bacteria that can immediately colonize the gut and provide measurable health benefits to maintain productivity

and health in livestock, whereas the other compounds primarily act as indirect modulators or substrates, which may not guarantee the same consistent therapeutic outcomes [15]. Probiotics are defined as non-pathogenic live microorganisms which improve host health when administered in adequate amounts [16]. Lactic acid bacteria (LAB) including different species and strains of *Lactobacillus, Lactococcus, Streptococcus, Enterococcus and Pedicoccus* are commonly used potential probiotic microorganisms based on selection criteria [17,18]. Several previous reports have shown that LAB improve overall flock health and reduces the risk of enteric diseases, boosts growth, enhances immune function, and improves the efficiency of nutrient utilization in poultry [19,20]. The addition of LAB probiotics as a food supplement has been shown to mitigate heat stress-related diseases in chickens, enhance antibody production, and improve immunity through the modulation of Toll-like receptor (TLR) signaling [21,22]. Furthermore, metabolic components derived from probiotics, including bacteriocins, amines and hydrogen peroxide which have the ability to interact with particular targets within various metabolic pathways to enhance cell proliferation through the inhibition of regular apoptosis and the stimulation of cell differentiation [23]. It has been reported that adding probiotic supplements to broilers at a dosage of 0.5 g/kg improves the pH and texture of the meat [24].

Currently, there's a rising concern about the heavy use of antibiotics in poultry production, especially in developing countries like Bangladesh, leading to a growing demand for poultry probiotics [25]. However, the limited stability and variable performance of probiotic strains often hinder their widespread use [26]. Probiotic strains that are derived from their natural hosts tend to be more compatible with the gastrointestinal tract, allowing for better adherence, survivability, and efficacy than strains from other sources. Thus, the host-specific probiotic strains development is essential for maximizing health benefits and improving livestock production performance [27]. Therefore, the objective of this study is to isolate, identify and characterize LAB strains with excellent probiotic properties from the gastrointestinal tract (GIT) of retail broiler chickens for potential integration into poultry farming practices.

## 2. Materials and methods

The animal study protocol was approved by the Animal Welfare and Experimentation Ethics Committee, Bangladesh Agricultural University, Mymensingh-2202, Bangladesh (AWEEC/BAU/2025(2)/32(a), Date: 25-05-2025).

### 2.1. Isolation and phenotypic characterization of LAB

The GIT (intestines and cecum) of twenty broiler chickens was collected from different retail poultry shops in Mymensingh, Bangladesh (Table 1). The samples were maintained on ice during transportation to the laboratory and were promptly

Table 1. Geographic details of retail chicken samples shown by sampling locations, number of chickens collected, identification number and coordinates.

| SL No. | Location | Number of Chicken Collected | Bacterial Isolate No. | Coordinates |
|---|---|---|---|---|
| 1 | K.R. MARKET, BAU | 4 | MCI1, MCI2, MCI3, MCI4 MCC1, MCC2, MCC3, MCC4 | 24.72693, 90.43662 |
| 2 | Kewatkhali Bazar | 4 | MCI5, MCI6, MCI7, MCI8 MCC5, MCC6, MCC7, MCC8 | 24.73830, 90.42520 |
| 3 | Mechua Bazar | 4 | MCI9, MCI10, MCI11, MCI12 MCC9, MCC10, MCC11, MCC12 | 24.738506, 90.424664 |
| 4 | Shes More Bazar | 4 | MCI13, MCI14, MCI15, MCI16 MCC13, MCC14, MCC15, MCC16 | 24.71771, 90.44414 |
| 5 | Sutiakhali Bazar | 4 | MCI17, MCI18, MCI19, MCI20 MCC17, MCC18, MCC19, MCC20 | 24.697372 90.454762 |

Note: MCI-Mymensingh Chicken Intestine, MCC- Mymensingh Chicken Cecum.

washed using 0.9% sodium chloride (NaCl) solution upon arrival. After washing each section of GIT, a 5 g sample from each section was transferred to 20 ml of de Man, Rogosa, and Sharpe (MRS) broth (Hi-Media, India) for enrichment and incubated aerobically at 37°C for 24 hours. Cultures displaying visible growth were selected, streaked onto MRS agar plates, and incubated anaerobically at 37°C for 24–72 hours. Only colonies showing greyish and creamy white appearance (presumptive for LAB) were selected and undergo three consecutive transfers on MRS agar plates to purify individual colonies [28,29]. Pure cultures were characterized morphologically using Gram staining and analyzed biochemically with catalase tests and carbohydrate fermentation profiles (glucose, sucrose, lactose, maltose, fructose and d-mannitol) according to standard procedures [30].

## 2.2. Assays of probiotics

**2.2.1. Hemolytic activity.** Fresh bacterial cultures were streaked onto Columbia blood agar base (HiMedia India) plates supplemented with 5% (v/v) sheep blood. Following incubation at 37°C for 24 hours, the isolates were assessed for clear zones around the colonies. Clear, colorless/lightened yellow zones depicting total Red Blood Cell (RBC) lysis are considered as beta (β) hemolysis. Alpha (α) hemolysis is characterized by a small greenish zone around the colonies due to the conversion of hemoglobin to methemoglobin, and gamma (γ) hemolysis refers to the absence of any hemolytic zones, indicating no hemolysis. Colonies displaying non hemolytic activity were selected, and those displaying beta or alpha hemolysis pattern were excluded [31].

**2.2.2. NaCl tolerance assay.** Salt tolerance of the bacterial strains was assessed by inoculating 50 µL of overnight-grown culture (1% v/v) into 5 mL of MRS broth supplemented with 2.0%, 4.0%, and 6.0% (w/v) NaCl (Sodium Chloride; Merck, India). After incubating for 24 hours at 37°C, bacterial growth was determined by measuring the optical density at 600 nm (OD$^{600}$) [32].

**2.2.3. Survival under low pH.** The acid resistance assay was conducted using the protocol described by Amelia et al. (2020) [33] with minor modifications. Briefly, the MRS broth medium was prepared at pH of 3 by using 1N HCl. The bacterial isolates were first cultured overnight in MRS broth under anaerobic conditions at 37°C. Then, 1 ml overnight bacterial culture was inoculated in 9 ml broth medium and incubated at 37°C for 3 h. Samples were collected at the beginning (0 h) and after 3 hours of incubation. These samples were serially diluted tenfold in sterile saline solution (5.8 g/L NaCl), and 1 mL of each dilution was plated on MRS agar. Plates were incubated aerobically at 37°C for 48–72 hours. The survival rate was measured by counting colony-forming units (CFU) at both 0 and 3 hours, following the method described by Bazireh et al. (2020) [34].

**2.2.4. Tolerance to simulated gastric juice.** Overnight incubated MRS broth of LAB cultures was centrifuged (7000×g) for 10 min. The resulting bacterial pellets were washed twice with sterilized phosphate-buffered saline (PBS) (8.0 g/L NaCl, 0.2 g/L KH$_2$PO$_4$, and 1.15 g/L Na$_2$HPO$_4$, pH 7.2), and a suspension containing $10^8$ CFU/ml was prepared. For each strain, 0.5 ml of this bacterial suspension was added to 4.5 ml of sterile artificial gastric juice media (125 mM NaCl, 7 mM KCl, 45 mM NaHCO$_3$, 3 g/L pepsin (Sisco Research Laboratories Pvt. Ltd. India), adjusted to pH 3.0 and sterilization by filtration) and incubated at 37 °C for 3 hours. Samples were collected at 0 h and 3 h after incubation and subsequently plated on MRS agar for the determination of viable cell counts [35].

**2.2.5. Tolerance to bile salt.** Bile salt tolerance was assessed using a slightly modified version of the method previously described by Naim et al. (2019) [36]. In brief, overnight bacterial cultures were inoculated (1% v/v) into two media: MRS broth containing 0.3% (w/v) ox-bile salts and regular MRS broth without bile salts as a control. Both samples were incubated at 37°C, and OD$^{600}$ was measured using a spectrophotometer at 0 h and 4 h after incubation. Subsequently, the percentage of growth suppression was measured by using the following formula:

$$\% \text{ of suppression} = \frac{\text{Growth in control} - \text{Growth in bile salt}}{\text{Growth in control}} \times 100$$

**2.2.6 Antimicrobial activity.** Antimicrobial activity assays were carried out using the well diffusion method with pathogenic bacteria *Escherichia coli*, Methicillin-resistant *Staphylococcus aureus* (MRSA), *Enterococcus faecalis*, *Salmonella typhimurium* and *Klebsiella pneumoniae*. These pathogenic organisms were obtained from the Microbiology Laboratory, Department of Microbiology and Hygiene, Bangladesh Agricultural University. Overnight grown pathogenic bacterial cultures were diluted in Buffered Peptone Water (BPW), and 200 μL ($10^7$ CFU/ml) of each bacterium was spread on Mueller-Hinton agar plates before wells were cut. LAB isolates were grown overnight in MRS broth at $37^0$C. Then, cell-free supernatants (CFS) were prepared by centrifugation of LAB cultures at 6000 rpm for 10 minutes and 100 μL ($10^8$ CFU/ml) aliquots were deposited into each well. After 24 h of incubation at $37^0$C, antimicrobial activity of the gut associated bacteria was determined by measuring the diameter of clear zones of inhibition around the wells using a transparent ruler [36].

**2.2.7. Antibiotic sensitivity.** The antibiotic resistance profile of LAB isolates was assessed on MRS agar using the antibiotic disc diffusion method. The agar medium was poured into plates and allowed to solidify at room temperature. The overnight-grown LAB cultures were uniformly spread over the solidified agar surface using a sterile spreader and then allowed to dry. Once the cultures were evenly spread, 14 antibiotic disks were placed on the surface of the agar plate. Subsequently, the plates were incubated at 37°C for 24 hours. The antibiotic susceptibility pattern of the isolates was assessed using nalidixic acid (30 μg/disc), ciprofloxacin (5 μg/disc), chloramphenicol (30 μg/disc), cefepime (30 μg/disc), imipenem (10 μg/disc), streptomycin 300, gentamicin (10 μg/disc), ampicillin (10 μg/disc), cefoxitin (30 μg/disc), sulfamethoxazole (25 μg/disc), cefotaxime (30 μg/disc), oxytetracycline (30 μg/disc), ceftazidime (30 μg/disc), and azithromycin (30 μg/disc). Following incubation, the diameter of inhibition zones around each disk was measured using a transparent ruler. Based on the measured diameters, inhibition zones were categorized as susceptible (≥21 mm), intermediate (16–20 mm), or resistant (≤15 mm) for each antibiotic [37].

**2.2.8. Cell adhesion assay.** The chicken ileum segment was placed in PBS at 4 °C for 30 minutes to remove surface mucus. The tissue was then cut into four segments (1 cm$^2$/1 cm$^2$). These segments were individually incubated to LAB bacterial suspensions ($10^8$ CFU/ml) at 37°C for varying durations: 0, 30, 60, and 90 minutes. Following incubation, the ileum segments were macerated to remove any bacteria that did not adhere. The processed samples were then plated onto Mueller-Hinton agar medium and cultured for 24 hours at 37°C. CFU were subsequently enumerated on each plate [29].

## 2.3. Molecular identification by 16S rRNA sequencing

The LAB strains that fulfilled the probiotic selection criteria underwent molecular identification via 16S rRNA gene sequencing. DNA of four isolates was extracted using TIANamp Bacteria DNA Kit (TIANGEN), followed by PCR amplification in 35 μl reactions containing 16 μl of OneTaq Quick-Load 2X Master Mix, 1 μl each of forward (27F: 5'-AGAGTT TGATCMTGGCTCAG-3') and reverse primers (1492R: 5'-TACGGYTACCTTGTTACGACTT-3') [29], 14 μl of DNase-free water, and 3 μl of DNA template. The thermal cycling protocol (BIORAD, United States) included initial denaturation at 95°C for 5 min, followed by 35 amplification cycles (94°C for 1 min, 60°C for 1.30 min, and 72°C for 2 min), with a final extension at 72°C for 7 min. After amplification, 5 μl of PCR products were analyzed by 1.0% (w/v) agarose gel electrophoresis and visualized under UV transillumination using Image Master (BIORAD, United States). PCR products of the expected size (1.5 kb) were purified and subjected to Sanger sequencing. The LAB strains were identified by sequence homology searches using the basic local alignment search tool (BLAST) against GenBank reference database. Thereafter, the sequences were registered in the NCBI (https://www.ncbi.nlm.nih.gov/genbank/) database and assigned with accession numbers. To analyze evolutionary relationships among isolates, phylogenetic trees were generated using MEGA version 12 [38], employing neighbor-joining statistical method, maximum likelihood substitution, and 1,000-repeat Bootstrap analysis [39].

## 2.4. Statistical analysis

Statistical analyses were conducted using SPSS version 26.0 (SPSS, Chicago, IL, United States), Microsoft Excel, and GraphPad Prism 8.0.2 (GraphPad Software, San Diego, CA, USA). Mean differences between treatments were analyzed using one-way ANOVA with significance determined at $P < 0.05$. All experiments were performed triplicate, and the data was expressed as means ± standard deviations (SD).

## 3. Results

### 3.1. Characterization and identification of isolates

In total, 40 bacterial colonies were randomly selected from the cultures of samples taken from the GIT of the 20 chickens. The initial 11 probable probiotic bacterial candidates were identified among 40 bacterial colonies based on their typical morphological appearance (5 small pinpointed, grayish white colonies and 6 small pinpointed, creamy white colonies), Gram-positive, catalase-negative, and cocci-shaped characteristics. Further identification through sugar fermentation profiling indicated that the isolates likely belonged to be *Enterococcus* and *Pedicoccus* species (S1 Fig and Table 2).

### 3.2. Assays of probiotics

**3.2.1. Hemolytic ability.** In terms of hemolytic activity, seven of the eleven isolates (MCI2, MCI7, MCI10, MCI11, MCC6, MCC10 and MCC12,) showed no change in the color of the media, indicating non-hemolytic activity (γ-hemolysis) (S1 Table). No zones appeared around the colonies (S2 Fig), confirming the absence of RBC lysis and indicating that they are considered safe and best. They were used for subsequent assays.

**3.2.2. NaCl tolerance.** All the isolates demonstrated the ability to grow at NaCl concentrations of 2%, 4%, and 6%, with a general decrease in $OD^{600}$ as NaCl concentration increased (Fig 1 and S2 Table). At 2%, 4%, and 6% NaCl, isolate MCC6 displayed the highest ($P < 0.05$) salt tolerance with $OD^{600}$ values of 1.688 ± 0.04, 1.539 ± 0.03, and 1.422 ± 0.04, respectively, while isolate MCI11 consistently exhibited the lowest ($P < 0.05$) $OD^{600}$ across all concentrations. For the higher concentrations of 6% NaCl, very weak growth was observed for MCI2 and MCI11, with $OD^{600}$ values 0.123 ± 0.02 and 0.078 ± 0.01 respectively, and they were excluded from further analysis due to their poor performance.

**3.2.3. Survival under low pH.** After analysis of NaCl tolerance, selected 5 isolates were examined to pH 3. The colony counts of potential acid-tolerant isolates were determined on MRS agar at pH 3 following 0 and 3 hours of incubation. All isolates were able to survive at pH 3.0 after 3h incubation except MCC10. Strains MCI7, MCI10, MCC6

**Table 2. Morphological and biochemical characteristics of selected isolates.**

| Strains | Cell Morphology | Gram | Catalase | Carbohydrate Fermentation Test | | | | | |
|---------|-----------------|------|----------|---------|---------|---------|--------|----------|------------|
| | | | | Glucose | Sucrose | Lactose | Maltose | Fructose | D-Mannitol |
| MCI2 | Small pinpointed, Grayish white and Cocci | + | − | + | − | + | + | + | − |
| MCI6 | Small pinpointed, Grayish white and Cocci | + | − | + | + | + | + | + | − |
| MCI7 | Small pinpointed, Grayish white and Cocci | + | − | + | + | + | + | + | + |
| MCI9 | Small pinpointed, Creamy white and Cocci | + | − | + | − | + | + | + | − |
| MCI10 | Small pinpointed, Creamy white and Cocci | + | − | + | + | + | + | + | + |
| MCI11 | Small pinpointed, Creamy white and Cocci | + | − | + | + | + | + | + | + |
| MCC5 | Small pinpointed, Grayish white and Cocci | + | − | + | + | + | + | + | + |
| MCC6 | Small pinpointed, Creamy white and Cocci | + | − | + | + | + | + | + | + |
| MCC10 | Small pinpointed, Creamy white and Cocci | + | − | + | + | + | + | + | + |
| MCC12 | Small pinpointed, Creamy white and Cocci | + | − | + | + | + | + | + | + |
| MCC15 | Small pinpointed, Grayish white and Cocci | + | − | + | − | + | + | + | − |

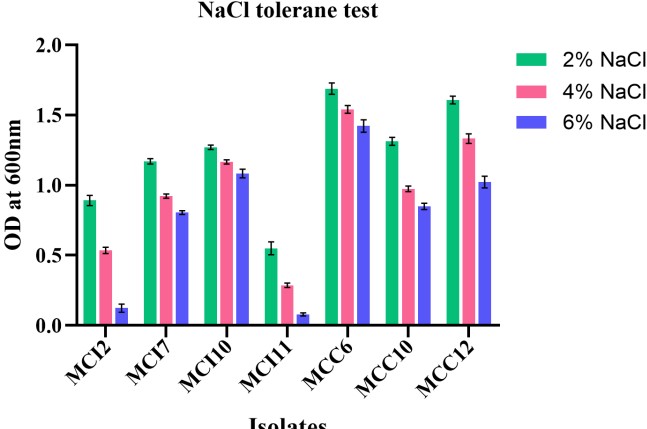

**Fig 1. Tolerance of selected LAB isolates to different concentrations of NaCl.** Growth of seven LAB isolates was assessed at 2%, 4%, and 6% NaCl by measuring optical density at 600 nm. The isolates exhibited strain-specific salt tolerance, with growth generally decreasing as NaCl concentration increased. Data are presented as mean±SD (n=3).

and MCC12 showed the relatively highest (P<0.05) survival rate ranging from 87.29 to 89.89% after 90 min of incubation, while MCC10 showed no viability excluded for further study. The results are shown in Table 3.

**3.2.4. Tolerance to simulated gastric juice.** The survivability of LAB isolates was examined in simulated gastric juice at pH 3.0. After 3 h of exposure to the simulated environment, the survivability of the four selected bacterial isolates decreased significantly (P<0.05). All isolates were able to survive with viable counts >3.0 CFU/ml (Fig 2). The highest (P<0.05) level of survival rates was observed for isolates MCI10 (59.06±1.15%) flowed by MCC6 (56.14±0.41%) and MCI7 (55.16±0.53%), while the lowest (P<0.05) survival level was observed for isolate MCC12 (51.01±0.82%).

**3.2.5. Bile salt tolerance.** After 4 h incubation at 37°C for the assessment of bile salt tolerance, isolates MCI10 and MCC6 showed the highest (P<0.05) resistance, with growth suppression 25.99±0.02% and 21.91±0.01% respectively. However, isolate MCI7 exhibited the most (P<0.05) suppression 32.78±0.02%, while MCC12 demonstrated 27.46±0.01%. In general, all the isolates exhibited bile salt tolerance above 50% (Fig 3). Additionally, MCC6 showed the greatest (P<0.05) OD$^{600}$, i.e., 0.438±0.01 in the presence of bile salts, while MCI7 had the lowest (P<0.05) value of 0.255±0.02 (S3 Table).

**3.2.6. Antimicrobial activity.** The antimicrobial activity of the isolated strains was evaluated against five types of pathogens, and the results are shown in Table 4 and S3 Fig. Using the agar well diffusion assay, selected 4 LAB strains (MCI7, MCI10, MCC6 and MCC12) showed no inhibition activity against *S. aureus* (MRSA), while MCI10 and MCC6

**Table 3. Survivability and acid tolerance of selected isolates in acidic conditions.**

| Isolates | pH tolerance (log CFU/ml) | | Survival rate (%) |
|---|---|---|---|
| | 0 h | 3 h | |
| MCI7 | 8.18±0.13 | 7.14±0.1 | 87.29 |
| MCI10 | 8.01±0.15 | 7.13±0.1 | 89.01 |
| MCC6 | 7.88±0.08 | 7.02±0.08 | 89.09 |
| MCC10 | 7.09±0.01 | ND | ND |
| MCC12 | 8.11±0.01 | 7.29±0.07 | 89.89 |

Note: ND – not determined; ± indicates standard deviation from the mean.

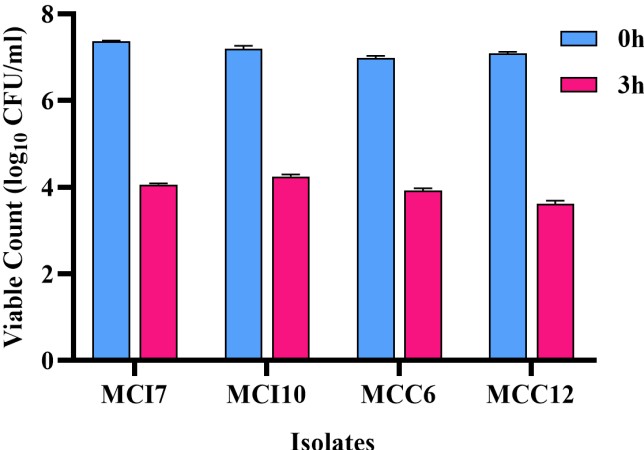

**Fig 2. Survival of lactic acid bacteria isolates in simulated gastric juice at pH 3.0.** Viable cell counts ($\log_{10}$ CFU/ml) of four selected LAB isolates (MCI7, MCI10, MCC6, and MCC12) were determined after 0 and 3 hours of incubation. Results are expressed as mean ± SD of triplicate experiments.

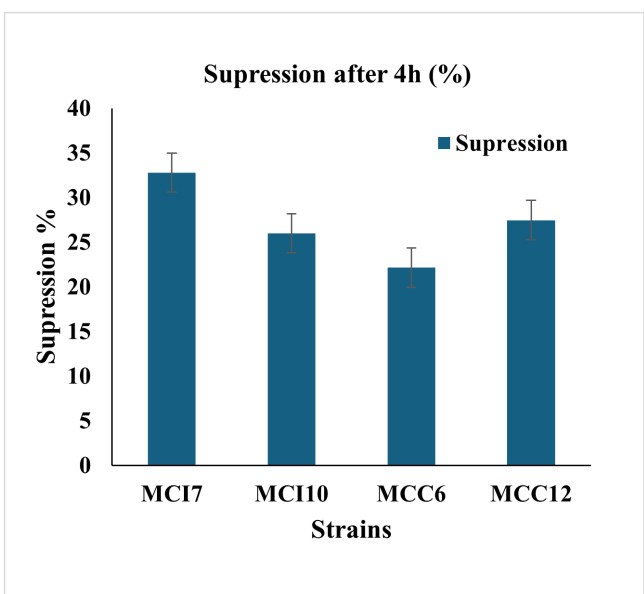

**Fig 3. Effect of bile salts on growth inhibition rates (suppression) of four selected isolates.**

strains were able to inhibit the *E. faecalis*. All strains demonstrated broad-spectrum activity, inhibiting three types of pathogens including *E. coli, S. typhimurium* and *K. pneumoniae*.

**3.2.7. Evaluation of antibiotic susceptibility.** The antibiotic susceptibility results of the four selected LAB isolates were tested against different antibiotics. All the isolates exhibited differing levels of antibiotic sensitivity and are presented in Table 5 and S4 Fig. The results indicated that all the strains were resistant to nalidixic acid, ciprofloxacin, cefoxitin, sulfamethoxazole and tetracycline, while each of the strains was susceptible to ciprofloxacin. No antibiotic resistance patterns were observed for cefepime, imipenem and cefotaxime except MCI7. Three of the four strains were effective inhibitors against gentamicin and azithromycin.

**Table 4. Antimicrobial activities against five pathogens.**

| Isolate No. | Diameter (mm) of inhibition zone | | | | |
|---|---|---|---|---|---|
| | *E. coli* | *S. aureus MRSA* | *E. faecalis* | *S. Typhimurium* | *K. pneumoniae* |
| MCI7 | 10 | – | – | 13 | 11.5 |
| MCI10 | 12 | – | 12 | 15.5 | 13 |
| MCC6 | 14.5 | – | 11 | 16.5 | 14 |
| MCC12 | 12 | – | – | 16 | 13 |

The zone diameter measurements represent the mean values from two experimental trials, expressed in millimeters (mm).

**Table 5. Susceptibility of potential LAB probiotic strains against some antibiotics measured by agar disc diffusion.**

| Strains | NA | CIP | C | FEP | IMP | S | CN | AM | FOX | SXT | CTX | T | CAZ | AZM |
|---|---|---|---|---|---|---|---|---|---|---|---|---|---|---|
| MCI7 | R | R | S | R | R | R | R | R | R | R | R | R | R | R |
| MCI10 | R | R | S | S | S | I | I | I | R | R | S | R | I | I |
| MCC6 | R | R | S | S | S | S | I | I | R | R | S | R | R | I |
| MCC12 | R | R | S | S | S | S | I | S | R | R | S | R | R | I |

R – Resistant; S -Sensitive; and I – Intermediate.

NA – nalidixic acid, CIP – ciprofloxacin, C – chloramphenicol, FEP – cefepime, IMP – imipenem, S – streptomycin, CN – gentamicin, AM – ampicillin, FOX – cefoxitin, SXT – sulfamethoxazole, CTX – cefotaxime, T – tetracycline, CAZ – ceftazidime and AZM – azithromycin.

### 3.2.8. Cell adhesion assay.

Effective probiotics must adhere to mucosal and epithelial surfaces to survive, persist, and potentially provide beneficial effects in the GIT. After 90 minutes of incubation, MCI10, MCC6, and MCC12 showed good adherence to chicken ileum epithelial cells, with viable counts ranging from $3.30\pm0.02$ to $3.40\pm0.02$ $Log_{10}$ CFU/ml, compared to MCI7, which showed a lower count of $2.99\pm0.07$ $Log_{10}$ CFU/ml (Fig 4). The results were not significantly different with $P>0.05$ of triplicate experiments.

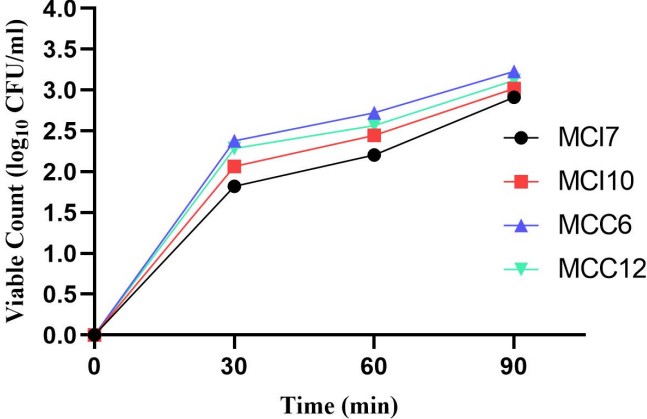

**Fig 4. Adhesion of lactic acid bacteria (LAB) isolates (MCI7, MCI10, MCC6, and MCC12) to poultry ileum epithelial cells, expressed as viable counts ($log_{10}$ CFU/ml) at 0, 30, 60, and 90 minutes of incubation.**

### 3.3. Molecular characterization

Molecular analysis was used to identify potential strains of LAB probiotics. All four LAB isolates displayed a band of 1500 bp in size after electrophoresis of the PCR amplification products on an agarose gel (Fig 5). Sequences of the 16S rRNA gene were analyzed using BLAST and subsequently deposited in the GenBank database, where accession numbers were assigned (Table 6).

### 3.4. Phylogeny analysis

A phylogenetic tree (Fig 6) was constructed using the neighbor-joining method based on evolutionary distances derived from the aligned 16S-rRNA sequences. Based on phylogenetic analyses, MCI10 and MCC12 isolates clustered into the clade of *P. pentosaceus* strains SV1 and SC2943 with moderate bootstrap support (57–66%). Similarly, isolate MCC6 grouped into the cluster of other *P. pentosaceus* strains (96–100% bootstrap), confirming its taxonomic placement within the species. MCI7 isolate clustered strongly with *E. faecium* strains (96–100% bootstrap), indicating genetic similarity within the species.

## 4. Discussion

In recent years, probiotics have gained considerable attention as promising alternatives to growth-promoting antibiotics in poultry production, while mitigating concerns about antimicrobial resistance and food safety [40]. Most probiotic strains are

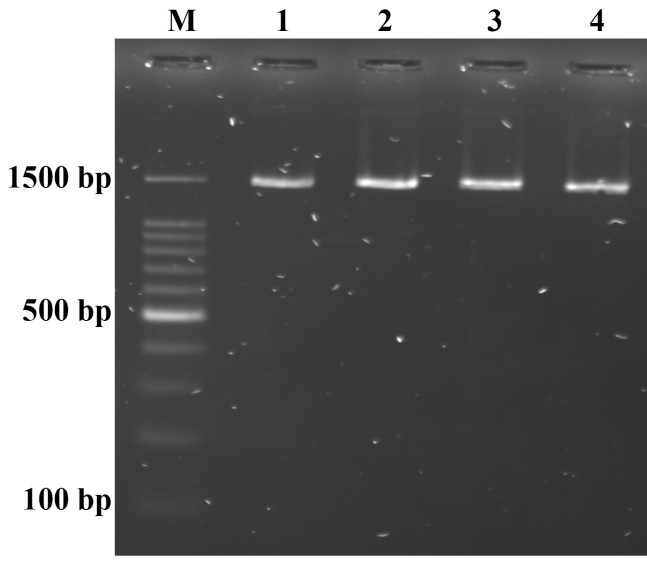

**Fig 5. Electrophoretic analysis of amplified 16S rRNA PCR products using agarose gel: lane M, 100 bp Marker (ADDBIO, SOUTH KOREA), Lanes 1, 2, 3, and 4 are positive lactic acid bacteria (LAB) strains MCI7, MCI10, MCC6 and MCC12 at 1500 bp.**

**Table 6. Species-level identification of probiotic LAB isolates using 16s rRNA sequencing.**

| Strains | Sequence Accession Number | Sequence Identity % |
|---|---|---|
| *Enterococcus faecium* strain MCI7 | PQ819573 | 99.15 |
| *Pediococcus pentosaceus* strain MCI10 | PQ819578 | 98.97 |
| *Pediococcus pentosaceus* strain MCC6 | PQ819574 | 98.87 |
| *Pediococcus pentosaceus* strain MCC12 | PQ819579 | 99.16 |

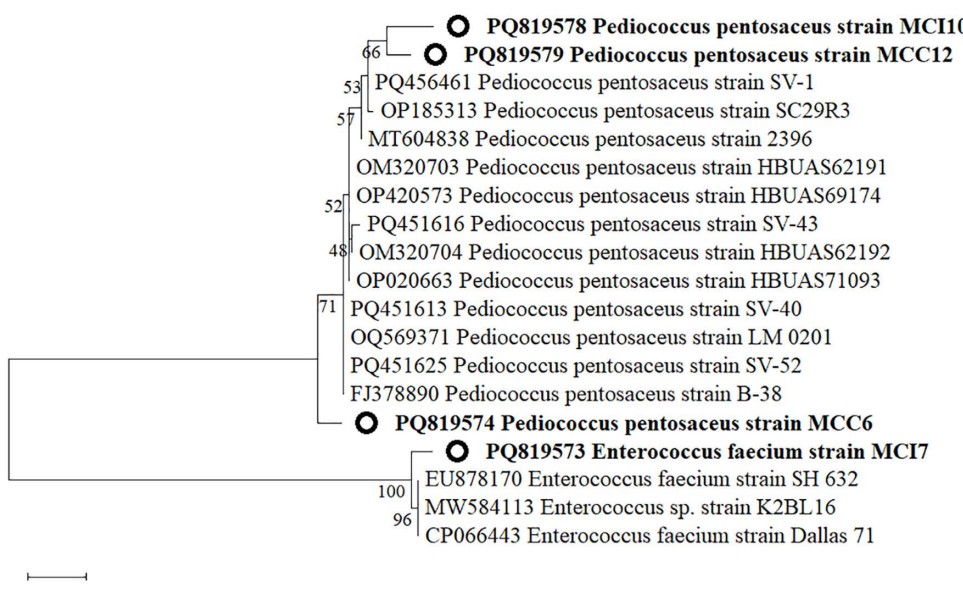

**Fig 6. Phylogenetic tree showing the relative positions of four isolates from the chicken gut and their related species.** The tree was constructed using the neighbor-joining method based on partial 16S rRNA gene sequences. Isolates obtained in this study are highlighted in bold, and the scale bar (0.01) represents the number of nucleotide substitutions per site.

isolated from the intestinal microbiota of humans and animals, as well as from fermented dairy products [41]. Host specific probiotics development is essential than strains isolated from the other sources because of their enhanced ability to colonize intestinal mucosa, allowing them to integrate more effectively into the host's gut microbiome [29]. Furthermore, direct *in vivo* evaluation of probiotics is often costly and time-consuming. As a result, *in vitro* testing is widely used to identify strains with the most beneficial properties. Additionally, evaluating the health effects of a particular probiotic strain in live animals can be very difficult and expensive [42]. In this study, LAB strains were isolated from healthy broiler chickens and evaluated *in vitro* for their potential as poultry probiotics.

In our study, we successfully isolated and characterized potential LAB from 20 healthy broiler chicken individuals collected from retail chicken shops in Mymensingh (Bangladesh). Out of 40 microbial isolates, 11 were phenotypically and biochemically identified to be probable probiotic candidates. Among these, at least four LAB strains were genetically confirmed as *Enterococcus faecium* (MCI7) and *Pediococcus pentosaceus* (MCI10, MCC6 and MCC12) species after subsequent probiotic assays.

The FAO recommends that microbial strain being considered for probiotic use must be safe to their host [43]. Hemolysis is a virulence factor frequently associated with pathogenic microorganisms that produce hemolysins to break down membrane phospholipids, leading to cell lysis [44]. For safety assurance, the selection of non-hemolytic strains is recommended, as this characteristic indicates an absence of virulence potential in probiotic candidates [45]. In our study, 7 out of 11 isolates showed γ- hemolytic or no hemolytic activity and were therefore chosen for further evaluation due to their safety as potential probiotics. Many researchers have reported similar results in the literature indicating that probiotics typically do not exhibit any hemolytic activity [33,46,47].

Our study revealed that the selected LAB isolates showed good tolerance to 2, 4 and 6% NaCl, Similarly, Reuben et al., (2019) [29] found that all isolates of LAB from GIT of chicken could survive at a concentration of 6.5% NaCl, and weak growth was reported at 10% NaCl. An earlier study showed that LAB isolated from milk and dairy products, meats, chicken faeces and other sources can survive in NaCl concentrations ranging from 1.0% to 9.0% [48,49]. These results

underscore the promising characteristics of potential LAB strains as probiotics, such as promoting bacterial growth and producing valuable metabolites, with significant implications for food preservation and production.

The selection of good probiotic bacteria depends on their ability to survive low pH conditions, pass through the digestive tract, and successfully colonize the host's gut [50]. In our study, isolates MCI7, MCI10, MCC6 and MCC12 displayed high survival rates ranging from 87.29% to 89.89% at pH 3.0, indicating their potential to survive the harsh acidic conditions of the host gut. Similarly, our finding corresponds with previous studies showing that LAB strains from chickens exhibit moderate to good survival at pH 3.0 [51–53]. Furthermore, tolerance to simulated gastric juice represents an essential parameter for identifying promising probiotic candidates [54]. In the present research, four selected isolates were able to survive in simulated gastric juice at pH 3.0 after 3 h of incubation, the survival rates ranging from 51.01% to 59.61%. Similar to the results of the present study, LAB isolates isolated from indigenous chicken (*Gallus domesticus*) of Nepal showed survival at pH 3.0, with the survival ratio ranging from 1.2% to 62.1% [55]. Sim et al., (2018) [56] reported that *E. faecium* L11 isolated from chicken exhibited good tolerance to artificial simulated gastric juice, maintaining a viability of 66.8±3.3%, which was slightly higher than the results of this study. The findings of our study indicate that the ability of our potential LAB probiotic strains to survive in low pH and artificial simulated gastric juice conditions is strain specific.

Bile, a digestive fluid produced in the liver and stored in the gallbladder, can be toxic to certain microorganisms. To effectively colonize the gut, probiotics must be able to withstand and survive in the presence of bile [57]. The concentrations of bile salts in the duodenum and cecum of the chicken gastrointestinal tract are approximately 0.175% and 0.008% respectively [58]. However, numerous studies have taken into account the average intestinal bile concentration is 0.3% (w/v) in order to evaluate the bile salt tolerance of probiotic LAB strains [59]. In our study, all the selected LAB strains demonstrated moderate to good bile salt tolerance under 0.3% bile salt conditions after 4h incubation (S3 Table), indicating their ability to survive and grow in the gastrointestinal environment. Similarly, Shin et al., (2008) and Shokryazdan et al., (2014) [59,60], reported good tolerance to 0.3% bile salts by all LAB strains isolated from chicken.

Zoonotic diseases and foodborne bacterial pathogens can cause high mortality and morbidity in poultry, leading to substantial economic losses in the poultry industry [61]. One of the important criteria for selecting a good probiotic candidate is its ability to effectively inhibit these pathogenic bacteria [62]. LAB exhibits antagonistic activity through the secretion of various antimicrobial substances, including bacteriocins, organic acids such as hydrogen peroxide, lactic acid, diacetyl, and carbon dioxide [63]. In our study, the selected strains exhibited antagonistic activity against *E. coli, S. typhimurium* and *K. pneumoniae* while no antimicrobial activity was shown against *S. aureus* (MRSA). However, the inhibitory effects of MCI10 and MCI6 were found against *E. faecalis* (Table 4). Our findings align with numerous earlier studies showing that LAB isolated from poultry display a broad range of antagonistic effects against different pathogens [29,60,64]. In contrast, Kizerwetter-Swida and Binek [65] found that LAB from chicken intestinal tracts more effectively inhibit Gram-positive pathogens (including *Staphylococcus aureus* and *Clostridium perfringens*) compared to Gram-negative pathogens (including *Salmonella* and *E. coli*). However, de Almeida Júnior et al., (2015) [66] found that the inhibitory activity of LAB showed no relationship with whether the tested pathogens were Gram-positive or Gram-negative. Spanggaard et al., (2001) [67] previously observed that probiotics can inhibit the establishment of heterochthonous bacteria in the GIT by managing pathogens, indicating that probiotics could play a crucial role in pathogen control.

As a key feature, probiotic microbial strains should not possess or acquire any antibiotic-resistance genes, which could be transferred to enteric pathogens [61]. Therefore, the assessment of the antimicrobial susceptibility of isolated bacteria is crucial for selecting potential probiotic strains. The antibiotic sensitivity test clarified that isolates MCI10, MCC6 and MCC12 were resistant to cefoxitin, ciprofloxacin, nalidixic acid, sulfamethoxazole and tetracycline (Table 5). Reuben et al., [29] reported that LAB strains isolated from poultry gastrointestinal tract were resistant to ciprofloxacin and tetracycline. Also, several studies have evidenced the resistance to various antibiotics by LAB [68–71]. Isolate MCI7 was resistant to all tested antibiotics except chloramphenicol. A possible explanation for this result may be the horizontal transmission of antibiotic-resistant genes [72] and the overuse of antibiotics in Bangladesh.

Successful probiotic colonization depends on the ability of bacterial strains to attach to host intestinal epithelial surfaces [73]. The present investigation demonstrated that all bacterial isolates successfully attached to chicken ileum epithelial cells and maintained viability throughout the 90-minute incubation period, with viable counts ranging from $2.99 \pm 0.07$ to $3.40 \pm 0.02$ $\log_{10}$ CFU/ml. These results align with previous studies demonstrating that LAB isolates from chicken sources showed progressive improvement in adherence capabilities with increasing incubation time [29,47].

In the present study, four potential probiotic strains were identified, characterized, and genotypically assigned to the *Pediococcus* and *Enterococcus* genus. These native strains have unique properties that may make them useful in poultry industry. Further *in vivo* studies and clinical investigations are essential to determine optimal dosages, strain-specific benefits, and health-promoting effects for developing effective probiotic-based health interventions.

## 5. Conclusions

This research identified four potential probiotic bacteria from the gastrointestinal tracts of retail broiler chickens. The *in vitro* results indicated that, in particular, LAB strains *P. pentosaceus* MCI10, *P. pentosaceus* MCC6 and *P. pentosaceus* MCC12 showed relatively good antipathogenic activity, antibiotic sensitivity, low pH and bile salt tolerance, therefore, they could have the greatest potential for probiotic use in chickens as feed additives to combat pathogens, improving poultry health, enhancing productivity, and reducing antibiotic reliance. Further investigation is necessary to evaluate the health benefits and protective effects of these new probiotic candidates *in vivo*.

## Supporting information

**S1 Fig. Morphology and Gram stain results of four selected LAB probiotic strains: MCI7, MCI10, MCC6 and MCC12.** (A) Morphology results of MCI7; (B) Gram stain results of MCI7; (C) Morphology results of MCI10; (D) Gram stain results of MCI10; (E) Morphology results of MCC6; (F) Gram stain results of MCC6; (G) Morphology results of MCC12; and (H) Gram stain results of MCC12.
(TIFF)

**S2 Fig. Results of the hemolytic activity of the 4 selected lactic acid bacteria (LAB) probiotic strains from chicken GIT.** A) MCI7 B) MCI10 C) MCC6 and D) MCC12.
(TIF)

**S3 Fig. The inhibitory effects of isolated probiotic strains against pathogenic indicator bacteria.** The pathogenic indicator bacteria were (A) *E. coli*; (B) *S. aureus*; (C) *E. faecalis*; (D) *S. typhimurium*; and (E) *K. pneumonia*. In each MHA agar plate, (a) was added to the MRS broth as control; (b) was added to the cell-free supernatant of MCI7; (c) was added to the cell-free supernatant of MCI10; (d) was added to the cell-free supernatant of MCC6 and (e) was added to the cell-free supernatant of MCC12.
(TIFF)

**S4 Fig. Antibiotic susceptibility pattern of isolated probiotic strains against different antibiotics was determined by disk diffusion assay.** A) & B) *Enterococcus faecium* strain MCI7; C) & D) *Pediococcus pentosaceus* strain MCI10; E) & F) *Pediococcus pentosaceus* strain MCC6 and G) & H) *Pediococcus pentosaceus* strain MCC12.
(TIFF)

**S1 Table. Hemolytic activities of potential lactic acid bacteria (LAB) probiotic strains from chicken GIT.**
(DOCX)

**S2 Table. Survival of potential lactic acid bacteria (LAB) probiotic strains at 2, 4, and 6% NaCl concentration estimated through measuring OD$^{600}$.**
(DOCX)

**S3 Table. Bile salt resistance of potential lactic acid bacteria (LAB) probiotic strains, estimated by measuring OD$^{600}$ in the presence and absence of bile salts.**
(DOCX)

**S1 File. Raw images.**
(TIF)

## Acknowledgments

We acknowledge the staff of the Biotechnology Laboratory and the Zoonotic Disease Laboratory in Bangladesh Agricultural University for their logistical, and operational support throughout the study. We also express our gratitude to the members of the local community from whom we collected our samples.

## Author contributions

**Conceptualization:** Sangram Biswas, Azimun Nahar, Sabina Yasmin.

**Data curation:** Sangram Biswas, Lutfor Rahman, Md. Taofiqur Rahman.

**Formal analysis:** Sangram Biswas, Md. Taofiqur Rahman.

**Investigation:** Fahmida Khatun, Azimun Nahar, Sabina Yasmin.

**Methodology:** Sangram Biswas, Lutfor Rahman.

**Supervision:** Fahmida Khatun, Azimun Nahar, Sabina Yasmin.

**Validation:** Fahmida Khatun, Azimun Nahar, Sabina Yasmin.

**Visualization:** Sangram Biswas, Lutfor Rahman, Md. Taofiqur Rahman.

**Writing – original draft:** Sangram Biswas.

**Writing – review & editing:** Sangram Biswas, Lutfor Rahman, Susmita Chowdhury, Fahmida Khatun, Azimun Nahar, Sabina Yasmin.

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
