## [Decision Letter · Decision Letter 0]

23 Sep 2025

Dear Dr. Yasmin,

Thank you for submitting your manuscript to PLOS ONE. After careful consideration, we feel that it has merit but does not fully meet PLOS ONE’s publication criteria as it currently stands. Therefore, we invite you to submit a revised version of the manuscript that addresses the points raised during the review process.

We look forward to receiving your revised manuscript.

Kind regards,

Guadalupe Virginia Nevárez-Moorillón, Ph.D.

Academic Editor

PLOS ONE

Journal Requirements:

Reviewers' comments:

Reviewer's Responses to Questions

**Comments to the Author**

1. Is the manuscript technically sound, and do the data support the conclusions?

Reviewer #1: Yes

Reviewer #2: Yes

2. Has the statistical analysis been performed appropriately and rigorously?

Reviewer #1: No

Reviewer #2: Yes

3. Have the authors made all data underlying the findings in their manuscript fully available?

Reviewer #1: Yes

Reviewer #2: No

4. Is the manuscript presented in an intelligible fashion and written in standard English?

Reviewer #1: No

Reviewer #2: Yes

Reviewer #1: The short title title is the similar to the full title.

Line 145: Please, clarify that incubation was done aerobically.

Line 158-159: It is important to state the nature and possibly the source of these isolates. Are they typed or clinical isolates.

Line 161: Please, state the measure that was taken to standardize the pathogenic inoculum before spreading it. Also state the volume that was spread on the plate.

Line 230: Recast the caption.

Reviewer #2: Current study is dealing with the prediction of probiotic potential of chicken gut associated bacteria via in-vitro analysis. Although we cannot recommend these bacteria as feed supplements without in-vivo experimentation, however, this study is forming basis for future in-vivo experimentation. So, it can be useful in future and has novelty. However, the number of assays performed for analysis of probiotic characteristics are not sufficient. Some modifications are needed to address before considering this article for publication.

Minor comments:

Methodology

1. L106: write GIT not the full term.

2. L121 and 122 are not methodology. Shift to Results section.

3. Which carbohydrates were selected for fermentation potential. Please write the names of tests.

4. Shift L120 after L114.

5. L146: Write abbreviation of colony forming units.

6. L152: Write the optical density with abbreviation and later use only abbreviation.

7. L159: Write Typhimurium as typhimurium.

8. L169-170, rephrase the sentence.

9. L201-202: rephrase and write correctly.

10. L205: remove with from with 20 each from intestine and cecum.

11. L309: Among these not of these.

12. L309, mention 4 as four.

13. L322: Mention lactic acid bacteria as LAB.

Major comments:

Methodology:

1. You did not perform serial dilution of enriched cultures. How did you get the countable number of colonies in this procedure? Kindly show the pictures of plates obtained after streaking of enriched cultures on MRS agar plates.

2. Shift molecular identification after the Morphological and biochemical characterization.

3. L164: Don’t use the term “probiotics” for these bacteria rather use fish gut born bacteria or gut associated bacteria.

4. L173-176: Describe the concentrations of antibiotics used.

5. Also perform the analysis of bacteria growth in simulated gastric medium and intestinal cell adhesion assay to confirm its probiotic characteristics. If it is easy then also perform cholesterol assimilation assay. But you should perform at least first two assays. Because number of assays documented here are not enough to confirm the probiotic potential of bacteria.

Results:

1. Figure 1 legend is not satisfactory. Describe it in detail and comprehensive way.

2. Rephrase the Table 1 caption.

3. Mention the p value in main paragraph not as foot note.

4. L243: Growth inhibition capability is not a correct term. Kindly replace with suppression.

5. L285-290: phylogenetic analysis description is very general and poor. Please discuss relation of each isolate with related genera and species with reference to bootstrap values mentioning the reliability of relation.

6. Figure 4 legend is poorly written. Kindly consult with published articles to improve your figure legends and table captions.

Discussion

1. Don’t mention human microbiota as it is irrelevant.

2. L319: bacteria did not show excellent tolerance to 6% NaCl.

3. L320: As you did not detect any stress tolerance associated protein so it is irrelevant to mention this sentence. Remove it.

4. L334: No reason to compare with pigs gut associated probiotics. There is a huge volume of literature on chicken gut probiotics. Kindly cite the relevant work.

5. Please cite more work from literature to compare the current study results.

6. In conclusion, describe one to two sentences which type of experimentation is needed in future perspectives of this study.

**Do you want your identity to be public for this peer review?** For information about this choice, including consent withdrawal, please see our Privacy Policy

Reviewer #1: No

Reviewer #2: **Yes:**  Fatima Muccee

---

## [Author Response · Author response to Decision Letter 1]

4 Dec 2025

Editor-in-Chief

PLOS ONE

Dear Editor,

Re: Resubmission of Manuscript (Manuscript No: PONE-D-25-29338) Entitled, " In-vitro evaluation of probiotic potential of gut microbes isolated from retail chicken."

Thank you for the opportunity to revise and resubmit our manuscript. We appreciate the time and effort the reviewers and the editorial team have invested in evaluating our work. We have carefully considered each reviewer's comments and made the necessary revisions to address their concerns. Below, we provide a point-by-point response to each of the comments.

Additional Editor Comments:

Reviewer #1:

Comment: The short title is the similar to the full title.

Response: Short title has been changed “Isolation and characterization of chicken gut microbes for probiotic potential”.

Comment: Line 145: Please, clarify that incubation was done aerobically.

Response: Lactic acid bacteria (LAB) are facultative anaerobes capable of growing under both aerobic and anaerobic conditions. Genera such as Enterococcus, Pediococcus, and Lactococcus able to grow aerobically, and our Gram staining indicated cocci morphology. Therefore, plates were incubated aerobically in a standard incubator without sealing (no wrapping with Parafilm), allowing oxygen to diffuse freely through the Petri dish lid.

Comment: Line 158-159: It is important to state the nature and possibly the source of these isolates. Are they typed or clinical isolates.

Response: These pathogenic organisms were obtained from the Microbiology Laboratory, Department of Microbiology and Hygiene, Bangladesh Agricultural University (L168-170).

Comment: Line 161: Please, state the measure that was taken to standardize the pathogenic inoculum before spreading it. Also state the volume that was spread on the plate.

Response: Overnight grown pathogenic bacterial cultures were diluted in Buffered Peptone Water (BPW), and 200 μL (107 CFU/ml) of each bacterium was spread on Mueller-Hinton agar plates before wells were cut. LAB isolates were grown overnight in MRS broth at 370C (L170-172).

Comment: Line 230: Recast the caption.

Response: Fig 1. Tolerance of selected LAB isolates to different concentrations of NaCl. Growth of seven LAB isolates was assessed at 2%, 4%, and 6% NaCl by measuring optical density at 600 nm. The isolates exhibited strain-specific salt tolerance, with growth generally decreasing as NaCl concentration increased. Data are presented as mean ± SD (n = 3) (L245-248).

Reviewer #2:

Minor comments:

Methodology

1. Comment: L106: write GIT not the full term.

Response: GIT has been included (L108)

2. Comment: L121 and 122 are not methodology. Shift to Results section.

Response: Shifted

3. Comment: Which carbohydrates were selected for fermentation potential. Please write the names of tests.

Response: Carbohydrate fermentation (glucose, sucrose, lactose, maltose, fructose and d-mannitol) has been mentioned (L118).

4. Comment: Shift L120 after L114.

Response: Corrected (L116-119)

5. Comment: L146: Write abbreviation of colony forming units.

Response: Abbreviation of colony forming units (CFU) has been included.

6. Comment: L152: Write the optical density with abbreviation and later use only abbreviation.

Response: Optical density’s abbreviation (OD600) has been mentioned in later.

7. Comment: L159: Write Typhimurium as typhimurium.

Response: Corrected (L168)

8. Comment: L169-170, rephrase the sentence.

Response: The sentence has been rephrased (L180-181)

9. Comment: L201-202: rephrase and write correctly.

Response: Corrected (L219-220)

10. Comment: L205: remove with from with 20 each from intestine and cecum.

Response: 20 each from intestine and cecum has been removed (L223).

11. Comment: L309: Among these not of these.

Response: Corrected (L346)

12. Comment: L309, mention 4 as four.

Response: four has been mentioned (L345)

13. Comment: L322: Mention lactic acid bacteria as LAB.

Response: LAB has been mentioned (L356)

Major comments:

Methodology:

1. Comment: You did not perform serial dilution of enriched cultures. How did you get the countable number of colonies in this procedure? Kindly show the pictures of plates obtained after streaking of enriched cultures on MRS agar plates.

Response: Initially, we performed serial dilutions from pure culture and obtained the expected CFU (10⁸), which was used for further analysis. In this case, MCC6 showed the expected CFU at the 6th dilution, where 0.1 mL of culture was spread on the medium, while MCI7, MCI10, and MCC7 showed the expected CFU at the 5th dilution.

2. Comment: Shift molecular identification after the Morphological and biochemical characterization.

Response: Initially, 11 isolates were selected from a total of 40 isolates based on morphological and biochemical characterization. Subsequent screening for hemolytic activity resulted in 7 non-hemolytic isolates. Further evaluation for NaCl and simulated gastric juice tolerance reduced the number to 4 isolates. Finally, bile salt tolerance, antimicrobial activity, and antibiotic sensitivity tests confirmed the selection of these same 4 isolates. Therefore, proceeding with molecular identification immediately after morphological and biochemical characterization would not be logical.

3. Comment: L164: Don’t use the term “probiotics” for these bacteria rather use fish gut born bacteria or gut associated bacteria.

Response: Gut associated bacteria has been included (L175)

4. Comment: L173-176: Describe the concentrations of antibiotics used.

Response: The concentrations of antibiotics have been mentioned in bracket (L184-187)

5. Comment: Also perform the analysis of bacteria growth in simulated gastric medium and intestinal cell adhesion assay to confirm its probiotic characteristics. If it is easy then also perform cholesterol assimilation assay. But you should perform at least first two assays. Because number of assays documented here are not enough to confirm the probiotic potential of bacteria.

Response: Simulated gastric medium (L148-156) and intestinal cell adhesion assay (L191-197) have been performed.

Results:

1. Comment: Figure 1 legend is not satisfactory. Describe it in detail and comprehensive way.

Response: We have revised the Figure 1 legend to provide a clearer and more comprehensive description (L 245-248)

2. Comment: Rephrase the Table 1 caption.

Response: We have revised the Table 1 legend to provide a clearer and more comprehensive description (L 120-121)

3. Comment: Mention the p value in main paragraph not as foot note.

Response: p value has been mentioned in main paragraph.

4. Comment: L243: Growth inhibition capability is not a correct term. Kindly replace with suppression.

Response: Suppression has been replaced with growth inhibition capability (L271).

5. Comment: L285-290: phylogenetic analysis description is very general and poor. Please discuss relation of each isolate with related genera and species with reference to bootstrap values mentioning the reliability of relation.

Response: We have revised the phylogenetic analysis and discussed relation of each isolate with related genera and species with reference to bootstrap values mentioning the reliability of relation (L320-326).

6. Comment: Figure 4 legend is poorly written. Kindly consult with published articles to improve your figure legends and table captions.

Response: Figure 4 replaced with Figure 6 and revised the figure legend to provide a clearer and more comprehensive description (L327-330).

Discussion

1. Comment: Don’t mention human microbiota as it is irrelevant.

Response: Citation and reference of human microbiota has been removed.

2. Comment: L319: bacteria did not show excellent tolerance to 6% NaCl.

Response: Corrected (L 356)

3. Comment: L320: As you did not detect any stress tolerance associated protein so it is irrelevant to mention this sentence. Remove it.

Response: The line has been removed.

4. Comment: L334: No reason to compare with pigs gut associated probiotics. There is a huge volume of literature on chicken gut probiotics. Kindly cite the relevant work.

Response: Pigs gut associated probiotics have been removed and included chicken gut probiotics associated literature.

5. Comment: Please cite more work from literature to compare the current study results.

Response: Current literature has been included

6. Comment: In conclusion, describe one to two sentences which type of experimentation is needed in future perspectives of this study.

Response: The future perspectives have been included in the conclusion section of discussion (L421-425).

Additional comments:

Q: 142 line- Why and how could plates for the growth of LAB be incubated aerobically?

Response: Lactic acid bacteria (LAB) are facultative anaerobes capable of growing under both aerobic and anaerobic conditions. Genera such as Enterococcus, Pediococcus, and Lactococcus able to grow aerobically, and our Gram staining indicated cocci morphology. Therefore, plates were incubated aerobically in a standard incubator without sealing (no wrapping with Parafilm), allowing oxygen to diffuse freely through the Petri dish lid.

Q: 160 line- What same condition? Do you mean MRS broth was used to grow pathogens because BPW is not a culture medium.

Response: Overnight grown pathogenic bacterial cultures were diluted in Buffered Peptone Water (BPW), and 200 μL (107 CFU/ml) of each bacterium was spread on Mueller-Hinton agar plates before wells were cut. LAB isolates were grown overnight in MRS broth at 370 C (L170-172).

We hope that current revisions are sufficient to make our manuscript suitable for publication in the PLOS ONE Journal and look forward to hearing from you at your earliest convenience.

Sincerely,

Sabina Yasmin

Professor

Institute of Biotechnology

Bangladesh Agricultural University

Mymensingh-2202, Bangladesh

---

## [Editor Report · Decision Letter 1]

30 Dec 2025

In-vitro evaluation of probiotic potential of gut microbes isolated from retail chicken.

PONE-D-25-29338R1

Dear Dr. Yasmin,

We’re pleased to inform you that your manuscript has been judged scientifically suitable for publication and will be formally accepted for publication once it meets all outstanding technical requirements.

Kind regards,

Guadalupe Virginia Nevárez-Moorillón, Ph.D.

Academic Editor

PLOS One
---

## [Editor Report · Acceptance letter]

PONE-D-25-29338R1

PLOS One

Dear Dr. Yasmin,

I'm pleased to inform you that your manuscript has been deemed suitable for publication in PLOS One. Congratulations! Your manuscript is now being handed over to our production team.

Kind regards,

on behalf of

Dr. Guadalupe Virginia Nevárez-Moorillón

Academic Editor

PLOS One